# Inhibiting Endothelin Receptors with Macitentan Strengthens the Bone Protective Action of RANKL Inhibition and Reduces Metastatic Dissemination in Osteosarcoma

**DOI:** 10.3390/cancers14071765

**Published:** 2022-03-30

**Authors:** Javier Muñoz-Garcia, Jorge William Vargas-Franco, Bénédicte Brounais-Le Royer, Denis Cochonneau, Jérôme Amiaud, Marie-Françoise Heymann, Dominique Heymann, Frédéric Lézot

**Affiliations:** 1Institut de Cancérologie de l’Ouest, F-44805 Saint-Herblain, France; javier.munoz@ico.unicancer.fr (J.M.-G.); denis.cochonneau@ico.unicancer.fr (D.C.); marie-francoise.heymann@ico.unicancer.fr (M.-F.H.); dominique.heymann@univ-nantes.fr (D.H.); 2Department of Basic Studies, Faculty of Odontology, University of Antioquia, Medellin 53-108, Colombia; jorge.vargas@udea.edu.co; 3Nantes Université, F-44000 Nantes, France; benedicte.brounais@univ-nantes.fr (B.B.-L.R.); jerome.amiaud@univ-nantes.fr (J.A.); 4Nantes Université, CNRS, US2B, UMR 6286, F-44322 Nantes, France; 5Department of Oncology and Metabolism, Medical School, University of Sheffield, Sheffield S10 2TN, UK; 6Sorbonne Université, INSERM, UMR933, Hôpital Trousseau (AP-HP), F-75012 Paris, France

**Keywords:** osteosarcoma, RANKL, endothelin, bone protection, metastases

## Abstract

**Simple Summary:**

The efficacy of current osteosarcoma therapy is diminished by two adverse events, namely resistance to chemotherapy and metastatic dissemination. In recent decades, research has been devoted to reducing these adverse events. Inhibiting bone resorption has shown promising effects on metastatic dissemination and tumor growth, with, however, the formation of significant tumoral mineralized tissue. Endothelin signaling is implicated in activating the cell that forms the mineralized tissues, consequently the impact of inhibiting it alone and in combination with the inhibition of bone resorption was evaluated using osteosarcoma models. The results obtained showed that inhibiting endothelin signaling significantly reduced the formation of mineralized tumor tissue concomitantly to metastatic dissemination without affecting sensitivity to chemotherapy. This inhibition appears to be a promising new therapeutic tool in the fight against osteosarcoma.

**Abstract:**

Current treatments for osteosarcoma, combining conventional polychemotherapy and surgery, make it possible to attain a five-year survival rate of 70% in affected individuals. The presence of chemoresistance and metastases significantly shorten the patient’s lifespan, making identification of new therapeutic tools essential. Inhibiting bone resorption has been shown to be an efficient adjuvant strategy impacting the metastatic dissemination of osteosarcoma, tumor growth, and associated bone destruction. Unfortunately, over-apposition of mineralized matrix by normal and tumoral osteoblasts was associated with this inhibition. Endothelin signaling is implicated in the functional differentiation of osteoblasts, raising the question of the potential value of inhibiting it alone, or in combination with bone resorption repression. Using mouse models of osteosarcoma, the impact of macitentan, an endothelin receptor inhibitor, was evaluated regarding tumor growth, metastatic dissemination, matrix over-apposition secondary to RANKL blockade, and safety when combined with chemotherapy. The results showed that macitentan has no impact on tumor growth or sensitivity to ifosfamide, but significantly reduces tumoral osteoid tissue formation and the metastatic capacity of the osteosarcoma. To conclude, macitentan appears to be a promising therapeutic adjuvant for osteosarcoma alone or associated with bone resorption inhibitors.

## 1. Introduction

Osteosarcoma is the main pediatric malignant primary bone tumor [1]. Conventional polychemotherapy combined with surgery (for reference the EURAMOS protocol [2]), make it possible to attain a five-year survival rate of 70% in affected individuals [1]. Unfortunately, if metastases are detected at the time of diagnosis or in the case of resistance to chemotherapy there is a drop in overall survival to 30% [3]. In recent decades, significant research efforts have been devoted to deciphering the mechanisms implicated in the occurrence of both metastatic dissemination and tumor escape from chemotherapy. It was, thus, suspected that bone resorption contributed to metastatic dissemination in osteosarcoma [4] in addition to its impact on the growth of the primary tumor through an amplification process called “vicious cycle” [4]. However, the exact role played by osteoclasts in osteosarcoma progression remains controversial. Therefore, inhibiting bone resorption has been proposed as a promising neo-adjuvant strategy for osteosarcoma with expected effects on tumor growth and metastatic dissemination, but also on peritumoral bone erosion. This last effect may make it possible to reduce the size of the tissues removed during surgery (with respect to margins) and to facilitate the ensuing functional reconstruction. Several clinical trials assessing the most efficient inhibitors of bone resorption, zoledronic acid (Zometa: NCT00470223, NCT00691236, NCT00742924, NCT02508038, NCT02517918, and NCT03932071) and denosumab, a RANKL blocking antibody (NCT02470091), showed strong protection of peritumoral bone [5].

We previously showed the therapeutic benefits of RANKL blockade in preclinical models of osteosarcoma [6]. In these models, RANKL inhibition slowed down osteosarcoma development through the inhibition of bone resorption (“extrinsic” activity), as well as via a direct impact on RANK expressing cancer cells (“intrinsic” activity). In addition, we demonstrated that the RANK/OPG ratio in tumor cells was associated with their metastatic potency [6]. Bone resorption is important for skeleton growth, and RANKL blockade during childhood results in growth arrest [7]. Similar data have been reported with zoledronic acid [8,9,10,11]. Interestingly, this bone growth arrest was rapidly reversed after the end of treatment with the RANKL inhibitor, in contrast to the tooth eruption process, which remained definitively blocked for certain teeth [7,12,13]. Bone resorption blockade results in an enlargement of the trabecular bone formed by osteoblasts associated with the endochondral process. In addition, stiffening of periosteal and flat bones has also been reported, as observed in osteopetrosis [14]. Such swellings have an irremediable impact on bone architecture and its related mineral content which may persist long-term after the end of the treatment [10]. In this context, a blockade of bone apposition in parallel to inhibiting bone resorption may be a promising therapeutic strategy for osteosarcoma composed of osteoid-forming cells.

The endothelin system, composed of three ligands (ET1-3) and two receptors (ETA-B), has been implicated in several skeleton developmental processes, including morphogenetic field determinism and bone formation (see review [15]). Osteoblasts are the targets of this system with a major impact on their differentiation and function [16,17,18]. Powerful inhibitors of this system have been developed by targeting solely one or both receptors, such as the dual receptor inhibitor macitentan [19,20].

The aim of the present work is to evaluate the therapeutic benefits of inhibiting the endothelin system alone or in combination with RANKL blockade as adjuvant therapy in a murine preclinical model of osteosarcoma. Its potential effects on osteoid tissue formation, metastatic dissemination, and sensitivity to conventional chemotherapeutic agents were investigated. The present results support using macitentan in combination with RANKL inhibition as an adjuvant therapy for treating osteosarcoma, with a bone protective action and a significant decrease in the lung metastatic process.

## 2. Materials and Methods

### 2.1. Cell Culture

The murine osteosarcoma MOS-J cell line, called MOS-J/Native in this study, derived from a spontaneous C57BL/6 mouse osteosarcoma, was provided by Prof. L. Shultz [21]. Two subclones derived from this cell line were also used in the experiments [6]. The first clone, MOS-J/PG1, revealed a high proliferation rate in vitro in contrast to the second, MOS-J/A3N. These clones, as well as the MOS-J/Native, were grown in RPMI1640 medium (Lonza, Walkersville, MD, USA) supplemented with 5% fetal bovine serum (FBS; Hyclone, Logan, UT, USA) and a mix of 100 U/mL of penicillin and 100 µg/mL of streptomycin (Lonza). The human osteosarcoma cell lines G292 (clone A141B1), KHOS/NP (R-970-5) named HOS in the manuscript, 143B, MG63, SJSA-1, and SaOS2 were purchased at the American Type Culture Collection (ATCC, Manassas, VA, USA), respectively, under references CRL-1423, CRL-1554, CRL-8303, CRL-1427, CRL-2098, and HTB-85. The CAL-72 cell line was purchased at the DSMZ-German Collection of Microorganisms and Cell Cultures (DSMZ, Leibniz Institute, Braunschweig, Germany) under reference ACC-439. G292, HOS, 143B, MG63, SaOS2, and CAL-72 cell lines were cultured in DMEM (Lonza) and the SJSA-1 cell line in RPMI-1640 (Lonza), both supplemented with 10% FBS.

### 2.2. RNA Isolation, Reverse Transcription, and Quantitative PCR

Total RNA was extracted from the cell cultures cultured in 6-multiwell plates using Direct-Zol RNA miniprep (ZymoResearch, Irvine, CA, USA) following the manufacturer’s instructions. First-strand cDNA was synthesized starting from 3 µg of total RNA, using Maxima H Minus Reverse Transcriptase (Thermo Scientific, Waltham, MA, USA) and Random primers. Quantitative real-time PCRs were performed on the equivalent of 20ng of reverse-transcribed total RNA with 300 nM of each primer (Table 1) and SYBR Select Master Mix (Applied Biosystems, Foster City, CA, USA). The results were acquired and analyzed using the CFX96 real-time PCR detector system and its software (Bio-Rad, Marnes-la-Coquette, France). β-Actin was used as the internal control to calculate relative amplification.

### 2.3. In Vivo Mouse Models of Osteosarcoma

All procedures involving mice were conducted in accordance with the institutional guidelines of the French Ethical Committee (CEEA-6-PDL, agreement APAFlS#8449-20170 1 0316591455 v3). Mice were housed under pathogen-free conditions at the Experimental Therapy Unit of the Faculty of Medicine at Nantes University (Nantes, France). Concerning injection and tumor monitoring, mice were anesthetized by inhalation of isoflurane at 2% in air with a flow of 1 L/min. Tumor volume (V) was calculated twice a week using the formula: length × width × depth × 0.5. Data points were expressed as average tumor volume ± S.E.M. Mice were sacrificed as soon as the tumor volume reached 2500–3000 mm^3^ (10% of body weight) for ethical reasons. The number of macroscopic lung metastases was counted manually for each mouse on the whole lung by two independent investigators. Lung histologic sections were realized (HE staining) and automatically digitized with a NanoZoomer 2.0 RS (Hamamatsu Photonics, Shizuoka, Japan) to microscopically evaluate, using the NDP. View2 software, the metastases sizes through the measure of the covered surface (8 sections at different levels for each metastasis, 4 metastases by mouse, 8 mice by group). The MOS-J models were induced in 5-week-old female C57BL6/J mice (Janvier Labs, Le Genest Saint Isle, France) and *Rankl^-/-^* mice [22,23] by an intramuscular injection of 10^6^ cells. The HOS model was induced similarly by inoculating 2 × 10^6^ of HOS cells into 5-week-old female NMRI-Nude mice (Janvier Labs).

### 2.4. IK22-5 RANKL Blocking Antibody Injections

Mice were injected subcutaneously every three days with 75 µg of IK22.5 RANKL blocking antibody [24], starting 5 days after inoculation of the tumor cells and for five weeks as previously described [6]. Control animals were injected with physiologic serum.

### 2.5. Macitentan Treatment

The inhibitor of dual endothelin receptors A and B, macitentan (ACT-064992), was kindly provided by Actelion Pharmaceuticals LTD (Gewerbestrasse 16, CH-4123 Allschwil, Switzerland) under cover of a Material Transfer and Secrecy Agreement. Mice were treated orally at 10 or 30 mg/kg by daily gavage starting 5 days after inoculation with the tumor cells and for 5 weeks. Gavage of control animals was performed using the carrier solution.

### 2.6. Ifosfamide Treatment

Suboptimal treatment with ifosfamide (ASTA Medica Laboratories, Merignac, France) was obtained with intraperitoneal injections for 3 consecutive days at a dose of 10 mg/kg starting 5 days after inoculation of the tumor cells. Control animals were injected with the carrier solution. Suboptimal dose was used to be able to visualize either a sensitizing or a desensitizing effect of the other drugs.

### 2.7. Micro-CT Analysis

Analyses of bone microarchitecture were performed using a Skyscan 1076 in vivo micro-CT scanner (Skyscan, Kontich, Belgium). After sacrificing the mice, tests were performed on tibias for each group. All tibias were scanned using the same parameters (pixel size 9 µm, 50 kV, 0.5 mm Al filter, 16 min of scanning). The 3D reconstructions were performed using NRecon and CTvox software (Skyscan).

### 2.8. Histology

After euthanasia, samples were preserved and fixed in 4% of PFA, decalcified (bone samples) with 4.13% of EDTA and 0.2% of PFA in PBS using a microwave tissue processor (KOS, Milestone, Kalamazoo, MI, USA) for 4 days, and embedded in paraffin (all chemical products from Sigma Chemical Co., St. Louis, MO, USA). Classical Masson trichroma staining or Hematoxylin-Eosin (HE) staining were performed on 3 µm-thick sections. Measures of the trabecular bone thickness were realized microscopically on sections automatically digitized with a NanoZoomer 2.0 RS (Hamamatsu Photonics) using the NDP.View2 software (6 sections at different levels for each tibia, 8 mice by group).

### 2.9. Statistics

The differences between the experimental conditions were assessed with Student’s t test or a one-way ANOVA followed by the Mann–Whitney test. The results are given as a mean ± SEM or SD from at least three independent experiments. Results were considered significant at p-values of <0.05. GraphPad Prism 6 software (GraphPad Software, San Diego, CA, USA) was used for all statistical analyses.

## 3. Results

### 3.1. Correlation between the Aggressivity of the MOS-J Osteosarcoma Model and the Relative Expression of ET1 and ETA

The MOS-J cell line (MOS-J/Native) and the subclones A3N (MOS-J/A3N) and PG1 (MOS-J/PG1), when injected into the tibia vicinity of syngeneic C57BL/6J mice, all induced the formation of bone tumors. However, the speed of the tumor formation, the tumor-associated bone resorption and the tumoral osteoid tissue formation were distinct for each subtype of MOS-J cells (Figure 1A,B). The MOS-J/PG1 model was very aggressive, with rapid growth, exacerbated bone resorption, and significant osteoid formation, whereas the MOS-J/A3N model was less aggressive (Figure 1A,B). Interestingly, when the expressions of endothelins (ET1-3) and their receptors (ETA-B) were evaluated at the transcriptional level using RT-qPCR, only ET1 and ETA were differentially expressed between the three MOS-J cell types (Figure 1C). ETA expression was higher in the PG1 cells, while ET1 was mainly detected in the A3N cells, denoting an apparent correlation between ETA versus ET1 expressions and the aggressiveness of the tumors formed. This correlation suggested that inhibiting the endothelin receptors might be of therapeutic interest for aggressive osteosarcomas highly expressing ETA. In order to challenge this assertion, the impact of the inhibitor of the dual endothelin receptors, macitentan, was evaluated on the MOS-J/PG1 model.

### 3.2. The Impact of Treatment with Macitentan on the Aggressive Osteosarcoma Model Increased When Associated with RANKL Blockade

The impact of macitentan at moderate (10 mg/kg/day) and high (30 mg/kg/day) doses on ETA-expressing aggressive osteosarcomas was evaluated using the MOS-J/PG1 model (Figure 2). Whatever the dose considered, macitentan treatment alone was unable to reduce tumor growth (Figure 2A), but tumoral osteoid tissue volume appeared slightly less significant (arrows in Figure 2B). This reduction may have two non-exclusive origins: (i) a direct decrease in osteoid tissue formation; or (ii) an increase in tumor-induced resorption. The number of osteoclasts was then evaluated on histological sections using TRAP staining (data not shown) but no difference was observed at the time of sacrifice.

To go further in the analyses of the impact of macitentan on tumoral osteoid tissue formation, macitentan treatment was combined with a powerful inhibitor of osteoclastogenesis, the IK22.5 RANKL blocking antibody. This combination had no significant impact on tumor growth (Figure 3A) but drastically reduced the osteoid tissue volume that was also in part diminished by the IK22.5 injections alone (Figure 3B). This impact was confirmed in *Rankl^-/-^* mice with significantly reduced tumoral osteoid tissue formation despite the absence of any significant effect on tumor growth (Figure 3D–E).

Histological views (Figure 3C) confirmed the reduction in osteoid tissue formation following treatment with macitentan, but, interestingly, also demonstrated that the enlargement of trabecular bone secondary to inhibiting the resorption induced by RANKL blockade was in part reversed by this treatment (Trabecular bone thickness: IK22.5-R 918 ± 87μm; IK22.5-L 896 ± 99μm; IK22.5 + Macitentan-R 495 ± 76μm; IK22.5 + Macitentan-L 632 ± 101μm). These results revealed that macitentan was capable of reducing the mineral tissue apposition associated with tumoral and healthy osteoblasts.

Similar experiments carried out with the human HOS osteosarcoma cell line, which expressed the highest ETA versus ET1 relative expressions of all the human osteosarcoma cell lines assessed (Figure 4A), showed the same results. No impact on tumor growth was observed (Figure 4B) whereas the combination of macitentan and IK22.5 significantly reduced tumoral osteoid tissue formation and trabecular bone enlargement (Figure 4C).

### 3.3. Macitentan Treatment and RANKL Blockade Alone or Combined Did Not Alter the Response to Conventional Chemotherapy

According to the idea that inhibiting endothelin-mediated signaling might also reduce tumoral neoangiogenesis, the question is raised of a potential decrease in the response to conventional chemotherapy that mainly targets the tumor through vascularization. To answer this question, macitentan and IK22.5 treatments alone or together were combined with ifosfamide at suboptimal doses using the MOS-J/PG1 model. Ifosfamide at a suboptimal dose made it possible to significantly reduce the growth of MOS-J/PG1 tumors (Figure 5A). Macitentan alone or combined with the IK22.5 antibody had no impact on the therapeutic efficacy of ifosfamide (Figure 5A), but both drugs kept their functional impact on bone resorption and tumoral osteoid tissue formation (Figure 5B,C).

Inhibiting endothelin signaling had no apparent impact on the chemotherapeutic response of the osteosarcoma. This observation is crucial for its potential use as an adjuvant therapy. However, the question of its possible effect on metastatic dissemination is raised. To respond this question, the lung metastatic foci of MOS-J/PG1 tumor-bearing mice treated or not with macitentan were quantified.

### 3.4. Macitentan at a High Dose Reduced the Number and Size of Lung Metastases

Macitentan at 10 mg/kg/day had no consequence on the number and size of the lung metastases (data not shown) but at the high dose of 30 mg/kg/day, the number and size (estimated on histological sections through the covered surface measurement) of the metastatic foci decreased compared to the untreated group (Figure 6; Control 0.0899 ± 0.0049 mm^2^ versus Macitentan 0.0647 ± 0.0059 mm^2^). These results encourage the use of macitentan as an adjuvant for osteosarcoma treatment.

## 4. Discussion

The aim of the present study was to evaluate in a mouse model the value of inhibiting the endothelin system with the dual receptor inhibitor, macitentan, as an adjuvant therapy for osteosarcoma. More specifically, this was tested in combination with inhibition of bone resorption achieved through blockade of RANKL. The effects of a daily oral treatment (gavage) with macitentan on tumor growth, osteoid tissue formation, metastatic dissemination, and the efficacy of the conventional chemotherapeutic agent (ifosfamide) were evaluated using a syngeneic model and an orthotopic model of osteosarcoma corresponding respectively to murine MOS-J cell inoculation in C57BL/6J mice and to human KHOS cell injection in Nude mice [25]. The expression pattern of endothelin receptors was analyzed with RT-qPCR and showed that a unique expression pattern was observed in all the human osteosarcoma cell lines, with only ET1 and ETA detectable (Figure 5). The HOS cell line was chosen for further analyses based on the higher ETA/ET1 ratio observed in this cell line, taking into account the fact that a correlation was observed in the various MOS-J cell lines between this ratio and tumor aggressiveness. This restricted expression of ET1 and ETA in the osteosarcoma was supported by several studies [26,27,28]. However, low expression of ETB has been reported in a few studies carried out with rat-derived ROS17/2.8 and UMR106 osteosarcoma cell lines [29,30,31,32,33]. Regardless, the dual endothelin receptor inhibitor, macitentan, was used in this study to totally block endothelin signaling in both tumor cells and cells in the tumor microenvironment.

Macitentan at 10 or 30 mg/kg/day used alone or in combination with the RANKL blocking antibody did not modify tumor progression. This result suggested that the endothelin system was not directly implicated in tumor cell proliferation. This assertion was supported by previous works on osteoblasts which showed that endothelins (more specifically ET1) stimulated bone formation [16,34,35,36,37] through control of osteoblast differentiation, and activation by regulating expression of various proteins implicated in bone mineralization: osteocalcin [30,38,39], osteopontin [38], bone sialoprotein [40], and alkaline phosphatase [28]. The bone phenotype associated with the conditional invalidation of ETA, driven by the osteocalcin promoter in mature osteoblasts (Cre-Lox system), was highly demonstrative regarding the implication of the endothelin system in the late stage of osteoblast differentiation, with a significant reduction in bone formation [39]. Treatment with macitentan alone had an effect on tumor osteoid tissue formation in the mouse osteosarcoma models used. In addition, the combined treatment of macitentan with the IK22.5 anti-RANKL antibody, made it possible to drastically reduce tumor osteoid tissue formation and the enlargement of both trabecular bone and periosteal bone secondary to inhibiting bone resorption. These results are in line with the use of macitentan as an adjuvant in the treatment of osteosarcoma.

Regarding the potential impact of macitentan treatment on osteosarcoma’s resistance to conventional chemotherapy, several studies have demonstrated that ET1 was associated with chemoresistance to doxorubicin and cisplatin [27,41,42,43], suggesting that macitentan treatment may reduce such resistance. In the present study, the impact of macitentan treatment on the efficacy of ifosfamide was evaluated using a suboptimal dose of this third chemotherapeutic agent. No variation in its efficacy was reported, strengthening the therapeutic value of macitentan as an adjuvant agent for treating osteosarcoma.

The endothelin system (ET1-ETA axis) was previously shown to favor the migration and invasion of osteosarcoma cells [27,42,44,45], with a clear link to the occurrence of metastases [46,47]. Moreover, experimental ETA invalidation in the osteosarcoma cell was capable of blocking the metastatic process [47]. In that respect, the present work has established that the blockade of endothelin signaling by macitentan was capable of significantly reducing the number and size of lung metastases developed in the syngeneic model of osteosarcoma used. These results are again in agreement with the use of macitentan as an adjuvant for osteosarcoma treatment.

## 5. Conclusions

The present study evaluates the value of inhibiting the endothelin system with a dual endothelin receptor inhibitor in the treatment of aggressive ETA-expressing osteosarcoma. Our results show that despite the absence of effect on tumor growth, inhibiting the endothelin system was efficient in reducing tumor osteoid tissue formation and the bone apposition secondary to the bone resorption blockade achieved using an anti-RANKL antibody. Furthermore, this treatment did not affect the sensitivity to chemotherapy and, interestingly, a significant reduction in the number and size of the lung metastases was observed. To conclude, all the data collected argue in favor of using a dual endothelin receptor inhibitor in combination with RANKL blockade as an adjuvant therapy in osteosarcoma.

## Figures and Tables

**Figure 1 cancers-14-01765-f001:**
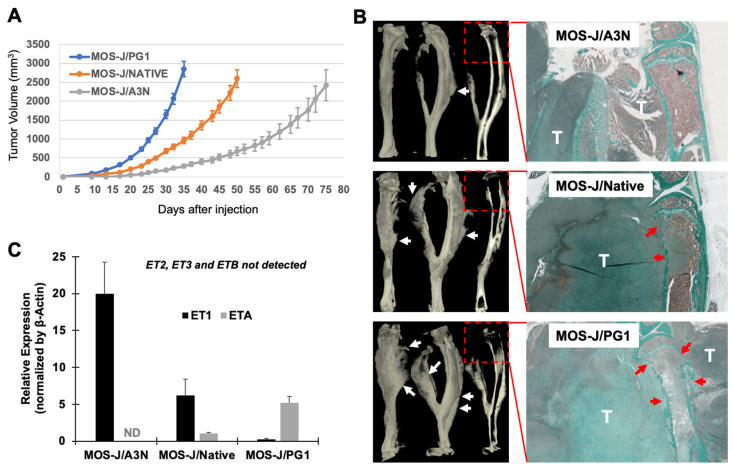
Comparative analyses of tumor growth, associated bone resorption and tumoral osteoid tissue formation between the three MOS-J osteosarcoma in vivo models and their relative expressions of ET1 and ETA. The MOS-J/PG1 cells showed a highly aggressive profile in contrast to the MOS-J/A3N cells, which were less aggressive in terms of tumor growth (**A**), bone resorption (red arrows in the histological views of (**B**)), and tumoral osteoid tissue formation (white arrows on micro-CT views of (**B**)). RT-qPCR analyses demonstrated that only ET1 and ETA were significantly detected in these cultures of MOS-J cell lines (A3N, Native, and PG1). ETA was highly expressed in the most aggressive cells and on the contrary ET1 in the less aggressive cells (**C**). T: tumor; and ND: not detected.

**Figure 2 cancers-14-01765-f002:**
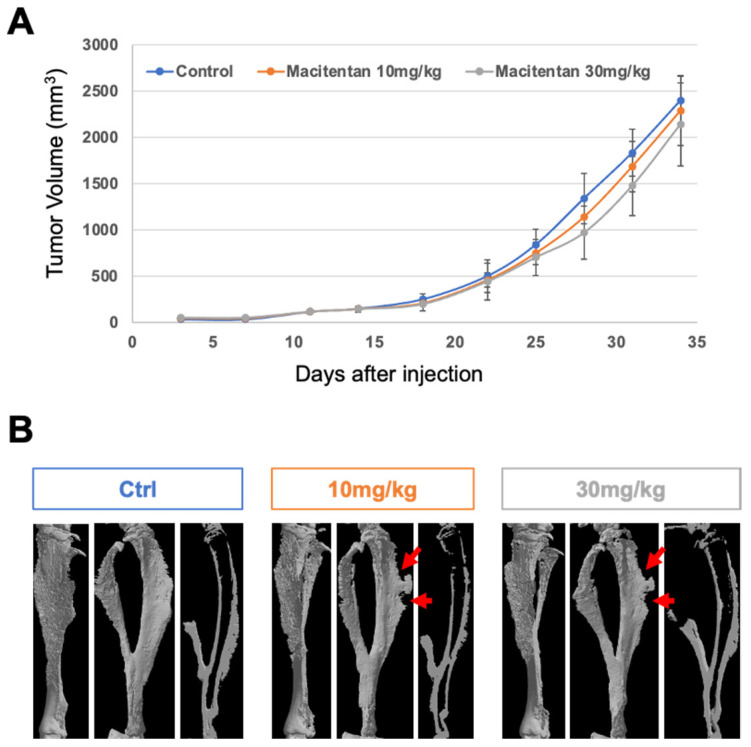
Impact of macitentan treatment alone at 10 and 30 mg/kg/day on MOS-J/PG1 tumor growth, associated bone resorption, and tumoral osteoid tissue formation. Whatever the dosage used, no significant impact of macitentan treatment was observed on tumor growth (**A**) but the final osteoid tissue volume appeared less significant (arrows in (**B**)).

**Figure 3 cancers-14-01765-f003:**
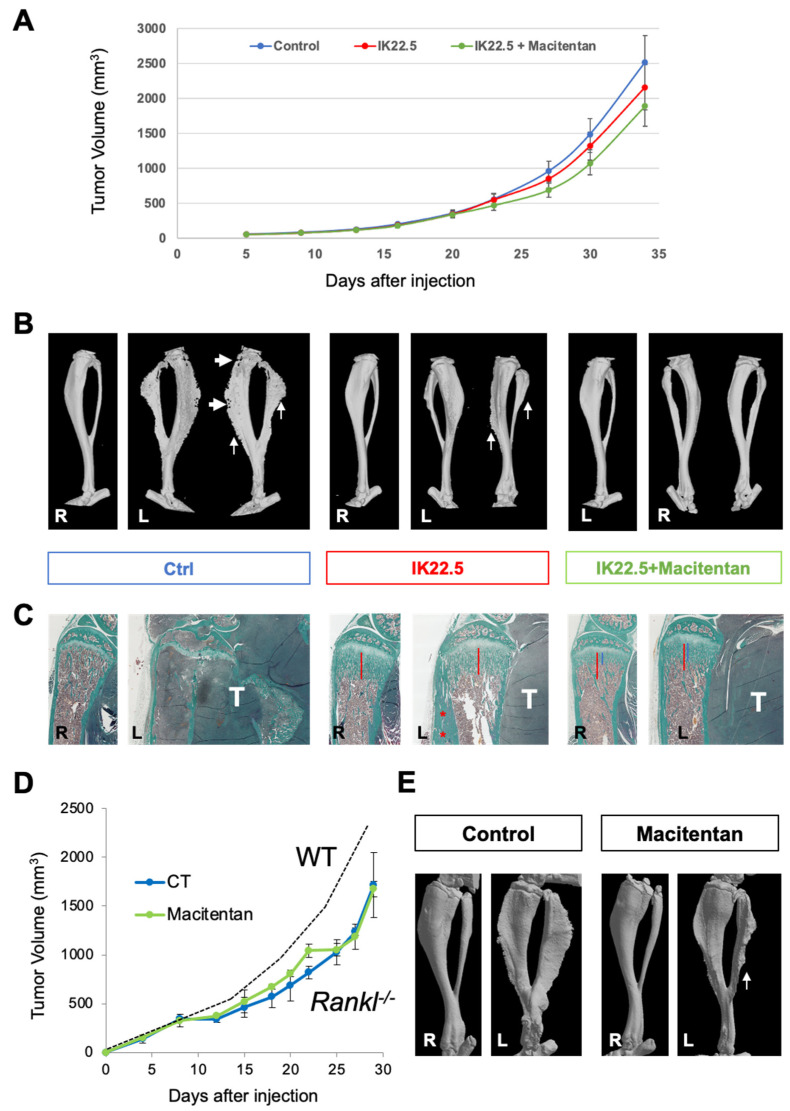
Macitentan (10mg/kg/day) strengthened the bone protective effect of the RANKL blocking antibody (IK22.5) injections in WT mice and reduced tumoral osteoid tissue formation in *Rankl^-/-^* mice either injected with MOS-J/PG1 cells. In the aggressive MOS-J/PG1 osteosarcoma model, RANKL inhibition had no significant impact on tumor growth (**A**) but clearly protected the bone from resorption (white arrow-heads in (**B**)) and reduced tumoral osteoid tissue formation (white arrows in (**B**)). Nevertheless, this treatment induced enlargement in both trabecular bone (red line in (**C**)) and cortical bone (red stars in (**C**)). Combining macitentan treatment with RANKL inhibition made it possible to reduce this enlargement (blue line in (**C**)). The MOS-J/PG1 osteosarcoma model could be applied to *Rankl^-/-^* mice (C57BL/6J background) and slightly slower tumor growth was observed compared to WT mice (**D**). Macitentan had no impact on tumor growth in *Rankl^-/-^* mice (**D**) but a significant reduction in tumoral osteoid tissue formation was observed (arrow in (**E**)) compared to the untreated *Rankl^-/-^* control mice (**E**). R: right tibia without tumor; and L: left tibia with tumor.

**Figure 4 cancers-14-01765-f004:**
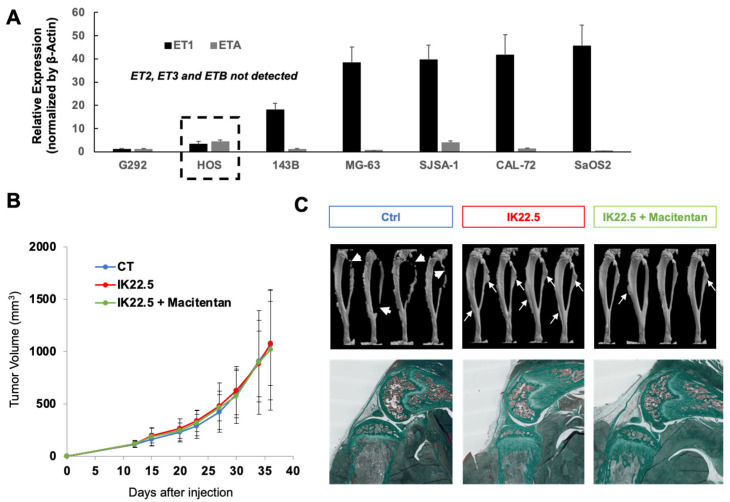
Macitentan strengthened the bone protective effect of RANKL blocking antibody (IK22.5) injections in the human HOS orthotopic mouse model. RT-qPCR analyses of endothelins and their receptor expressions in seven human osteosarcoma cell lines showed that only ET1 and ETA were detected, with a higher ETA expression compared to ET1 in the HOS cell line (**A**). Injections of IK22.5 RANKL blocking antibody alone or in combination with macitentan treatment, in the orthotopic osteosarcoma model corresponding to NRMI Nude mice injected with HOS cells, had no impact on tumor growth (**B**) but made it possible to protect bone from the tumor-induced resorption seen in untreated controls (arrow-heads in (**C**)). Adding a treatment with macitentan to IK22.5 injections also reduced tumoral osteoid tissue formation (arrows in (**C**)) and enlargement of the trabecular bone as seen in the histological sections (**C**).

**Figure 5 cancers-14-01765-f005:**
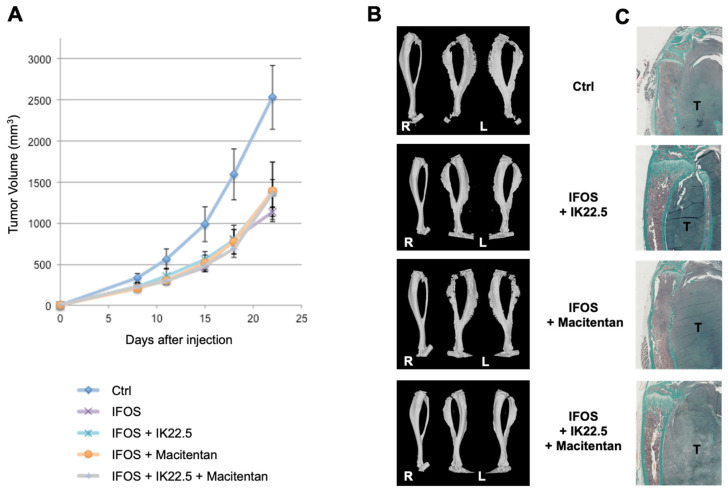
Macitentan and RANKL blocking antibody (IK22.5) injections alone or in combination did not interfere with the chemotherapeutic effect of ifosfamide, and combining them retained the major bone protective effect. Injections of a suboptimal dose of ifosfamide significantly reduced tumor growth in the MOS-J/PG1 mouse model of osteosarcoma (**A**). IK22.5 and macitentan treatment alone or in combination did not interfere with ifosfamide (**A**) but the combination still had a bone protective effect as seen in the micro-CT (**B**) and histological (**C**) views. R: right tibia without tumor; and L: left tibia with tumor; T: tumor.

**Figure 6 cancers-14-01765-f006:**
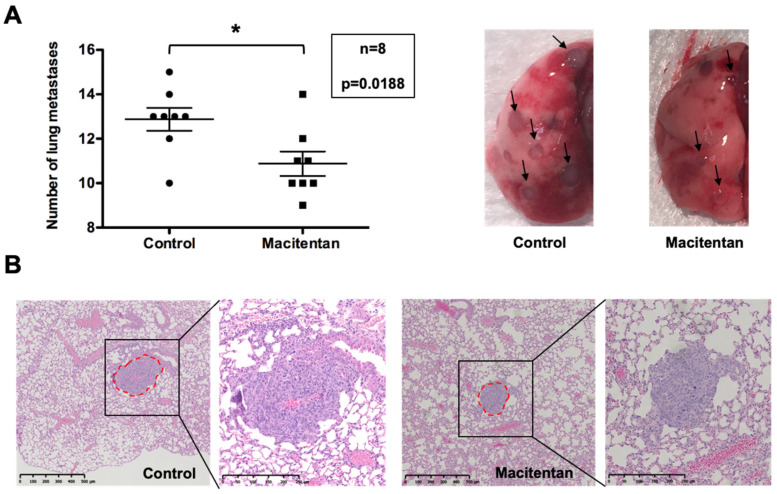
Macitentan treatment at a high dose reduced the number and size of lung metastatic foci. The number of metastatic nodules macroscopically detectable (arrows in (**A**)) in the lung of MOS-J/PG1 tumor-bearing mice reduced significantly in the group treated with 30 mg/kg/day of macitentan compared to the untreated control group (*p* = 0.0188) (**A**). Similarly, histological sections (HE staining) of the lungs showed lower sizes in the presence of macitentan (**B**). Red dotted line surrounding metastatic lung nodules in representative sections of each group (*n* = 8 mice per group) enabled to measure the covered surface. * means *p* < 0.05.

**Table 1 cancers-14-01765-t001:** Primers used for RT-qPCR.

**Mouse and Human**		
Primers	Sequences	Amplicon
ET1-Fw ET1-Rv	5′-ACT TCT GCC ACC TGG ACA TC-3′ 5′-CCA GCA CTT CTT GTC TTT TTG G-3′	142 pb
ETA-Fw ETA-Rv	5′-TAT TTT GTG AGC AAG AAA TT-3′ 5′-GGG GAC CGA GGT CAT-3′	55 pb
ETB-Fw ETB-Rv	5′-GGT CCC AAT ATC TTG ATC G-3′ 5′-CAA CAG CTC GAT ATC TGT CA-3′	171 pb
**Mouse**		
Primers	Sequences	Amplicon
ET2-Fw ET2-Rv	5′-CCT GGC TTG ACA AGG AAT GT-3′ 5′-CTT CGA TGG CAG AAG GTA GC-3′	181 pb
ET3-Fw ET3-Rv	5′-CCC TGG TGA GAG GAT TGT GT-3′ 5′-CTG GGA GCT TTC TGG AAC TG-3′	295 pb
βACT-Fw βACT -Rv	5′-CTA AGG CCA ACC GTG AAA AG-3′ 5′-ACC AGA GGC ATA CAG GGA CA-3′	140 pb
**Human**		
Primers	Sequences	Amplicon
ET2-Fw ET2-Rv	5′-GCT ATG GTC TCC GTG CCT AC-3′ 5′-GCC GTA AGG AGC TGT CTG TT-3′	243 pb
ET3-Fw ET3-Rv	5′-TCA ACA CTC CCG AAC AGA CG-3′ 5′-TGA CGT CCA GAG TTT GGG TG-3′	186 pb
βACT-Fw βACT -Rv	5′-CCT CGC CTT TGC CGA TCC-3′ 5′-AGG ATG CCT CTC TTG CTC TG -3′	243 pb

## Data Availability

The data presented in this study are available on request from the corresponding author.

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
