# Peer review of "Inhibiting Endothelin Receptors with Macitentan Strengthens the Bone Protective Action of RANKL Inhibition and Reduces Metastatic Dissemination in Osteosarcoma"

_cancers, 2022, doi:10.3390/cancers14071765_

Round 1

Reviewer 1 Report

The manuscript evaluates the value of inhibiting the endothelin system in the treatment of osteosarcoma. The study shows that inhibiting the endothelin system by macitentan was efficient in reducing tumor osteoid tissue formation and the number and size of lung metastases. In addition, the combined treatment of macitentan with the anti-RANKL antibody drastically reduced the tumor osteoid tissue volume and the enlargement of both trabecular bone and periosteal bone induced by RANKL blockade, indicating that the endothelin receptor inhibitor in combination with RANKL blockade could be as an adjuvant therapy in osteosarcoma. Overall, the manuscript is well written and the interpretation of data is generally sound. Here are some suggestions:

  1. The contents of Figure 4 can be embedded in Figure 3;
  2. Lines 317 to 321 on page 10 should be placed in the Result 3.3;
  3. It is recommended to place low and high magnification photos in the Figure 7B.

Author Response

Reply to Reviewer 1

Authors thank the Reviewer for the suggestions that have been all taken into account in the revised version of the manuscript. Indeed, the contents of Figure 4 has been embedded in Figure 3, lines 317 to 321 on page 10 have been placed in the Result 3.3 and low and high magnification photos introduced in the Figure 7B.

Authors hope that the revised form of the manuscript complies with all the Reviewer advices.

Reviewer 2 Report

The authors addressed an important topic - the need of finding new treatment strategies for metastatic osteosarcoma. The overall article is of interest to the readers and is well written. Nonetheless, several issues may have to be addressed or clarified before it can be considered for this journal.

Major points:

  • The experiment of Macitentan with RANKL antibody (figure 2) should include the crucial control of RANKL antibody only; otherwise, it is unclear how much of the effect on reduction of osteoid tissue can be attributed to Macitentan itself
  • The authors should explain how exactly tumor-induced resorption was measured; just showing photos is not enough and suggests a very subjective approach
  • Can the authors explain why "suboptimal" doses of IFA were used?
  • How did the authors measure the lung metastases macroscopically and why did they only show them microscopically instead? The authors should use both methods and demonstrate them in a figure and then clearly explain results and ranges in the results

Minor:

  • Overall, the methods should be more clear
  • The figures should be larger (some of the photos are very small and hard to interpret)

Author Response

Reply to Reviewer 2

Authors thank the Reviewer for all the comments that aim to improve the manuscript. Authors have taken into account all the comments and modified the manuscript in that way.

Here is a point-by-point answer to the Reviewer comments.

Comment: The experiment of Macitentan with RANKL antibody (figure 2) should include the crucial control of RANKL antibody only; otherwise, it is unclear how much of the effect on reduction of osteoid tissue can be attributed to Macitentan itself.

Response: The Figure 2 corresponds to Macitentan treatment alone at two different doses. The experiment of macitentan with RANKL antibody are presented in Figures 3 and 4 (5 in initial manuscript) and the control corresponding to only RANKL antibody (IK22.5) is always given as crucial.

Comment: The authors should explain how exactly tumor-induced resorption was measured; just showing photos is not enough and suggests a very subjective approach.

Response: Authors agree with the Reviewer that measuring the tumor-induced resorption is an important point that cannot be resumed to just showing photos. Regarding the effect of the RANKL blocking antibody, the total inhibition of the tumor-induced resorption was measured using microCT scan and already published in CANCERS (Navet et al., Cancers 2018, 10, 398; doi:10.3390/cancers10110398), that why we have chosen not to repeat here these data. Concerning the macitentan effect, we have observed no impact on the bone resorption, more precisely on the osteoclast number (TRAP staining) as written in 3.2. lines 268-269. The effect of macitentan is rather on the osteoblast lineage (“normal” and tumoral osteoblasts) inducing a reduction of the bone and tumoral-osteoid tissues formation. This is evidenced in figure 3 by the red and blue lines corresponding to the trabecular bone thickness in “tumoral” tibia (L) and contralateral “none-tumoral” tibia. (R). Trabecular bone thickness was measured on histologic sections automatically digitized with a NanoZoomer 2.0 RS (Hamamatsu Photonics, Shizuoka, Japan) using the NDP.View2 software, and results were added to the manuscript (line 290) to give an objective dimension to these results.

Comment: Can the authors explain why "suboptimal" doses of IFA were used?

Response: Authors thank the reviewer for asking a precision on that point. Suboptimal dose was used to be able to visualize either a sensitizing or a desensitizing effect of the other drugs. Indeed, IFA at optimal dose is very efficient on the MOS-J cells inducing rapidly more than 97% of cell-death. So, in order to be able to visualize potential effects of the other drugs we have chosen a suboptimal dose close from the IC50 of IFA on MOS-J cells. A sentence has been added to the materials and methods (2.6.).

Comment: How did the authors measure the lung metastases macroscopically and why did they only show them microscopically instead? The authors should use both methods and demonstrate them in a figure and then clearly explain results and ranges in the results.

Response: Authors thank the Reviewer for this comment. We agree that precisions were necessary. Macroscopic views of the lungs were added to the Figure with arrows evidencing the metastatic nodules. Macroscopic views were used to numbered the lung metastases (Figure 6A). The size of the metastatic nodules was evaluated on automatically digitized (NanoZoomer 2.0 RS) histologic sections (8 sections at different levels for each metastasis, 4 metastases by mouse, 8 mice by group) using NDP.View2 software to evaluate the surface covered by each nodule. Data were added to the text and technical precisions in the materials and methods.

Comment: Overall, the methods should be more clear.

Response: The methods has been amended in several parts. Authors hope this is clearer now.

Comment: The figures should be larger (some of the photos are very small and hard to interpret).

Response: Authors have added as possible some enlargements. The structure of the text with figures included (asked for the submission) reduces the size of the figures. High resolution figures have been submitted apart from the text but we don’t know if they are accessible to the reviewers during the review process!

Authors hope having answered to all comments of the Reviewer.

Round 2

Reviewer 2 Report

I think the authors addressed the concerns appropriately.